# JWSign: A Highly Multilingual Corpus of Bible Translations for more Diversity in Sign Language Processing

**Shester Gueuwou**[1], **Sophie Siake**[2], **Colin Leong**[3], **Mathias Müller**[4]

[1]Kwame Nkrumah University of Science and Technology, Ghana
[2]EFREI Paris, France
[3]University of Dayton, USA
[4]University of Zurich, Switzerland

## Abstract

Advancements in sign language processing have been hindered by a lack of sufficient data, impeding progress in recognition, translation, and production tasks. The absence of comprehensive sign language datasets across the world's sign languages has widened the gap in this field, resulting in a few sign languages being studied more than others, making this research area extremely skewed mostly towards sign languages from high-income countries. In this work we introduce a new large and highly multilingual dataset for sign language translation: JWSign. The dataset consists of 2,530 hours of Bible translations in 98 sign languages, featuring more than 1,500 individual signers. On this dataset, we report neural machine translation experiments. Apart from bilingual baseline systems, we also train multilingual systems, including some that take into account the typological relatedness of signed or spoken languages. Our experiments highlight that multilingual systems are superior to bilingual baselines, and that in higher-resource scenarios, clustering language pairs that are related improves translation quality.

## 1 Introduction

There are around 300 sign languages recorded up to date (United Nations, 2021). However, sign language translation research is extremely skewed towards a limited number of sign languages, primarily those from high-income countries (Müller et al., 2023), while ignoring the vast majority of sign languages used in low and middle-income countries (Gueuwou et al., 2023). A similar phenomenon was observed in the spoken[1] languages machine translation community and was shown by Ògúnremí et al. (2023) to be harmful, calling on the NLP community to do more research on low resource spoken languages (Ranathunga et al., 2023). This issue is exacerbated by the fact that approximately 80% of people with disabling hearing loss in the world reside in middle and low-income countries (World Health Organization, 2023).

Our first contribution towards addressing these challenges is to present *JWSign*, a highly multilingual corpus of Bible translations in 98 sign languages, made accessible through an automated loader. To the best of our knowledge, JWSign is one of the largest and most diverse datasets to date in sign language processing (§3).

There is precedent in natural language processing (NLP) for using Bible translations as a starting point for many under-resourced languages that may not have any parallel resources in other domains. Bible corpora have played a major role in research in speech and text areas of NLP (§2).

We complement the JWSign dataset with baseline experiments on machine translation, training a Transformer-based system for 36 bilingual pairs of languages in the dataset. Such bilingual systems, trained individually for each language pair (one sign language and one spoken language), are the default procedure in recent literature.

However, sign language translation (SLT) has proven to be a challenging task, due to obstacles such as very limited amounts of data, variation among individual signers and sub-optimal tokenization methods for sign language videos. A potential way to improve over bilingual systems (the predominant kind at the time of writing) is to build multilingual systems. Linguistic studies have suggested a good level of similarity and mutual intelligibility among some sign languages (Power et al., 2020; Reagan, 2021) even from different continents (e.g. Ghanaian Sign Language in Africa and American Sign Language in North America). In this work, we therefore explore different multilingual settings for sign language translation (§4).

---

[1]In this work, following Müller et al. (2022a), we "use the word 'spoken' to refer to any language that is not signed, no matter whether it is represented as text or audio, and no matter whether the discourse is formal (e.g. writing) or informal (e.g. dialogue)".

## 2  Related Work

The following section explores the motivation behind the research by examining works that have utilized the Bible in various modalities (§2.1). It delves into the limited coverage of many sign languages within popular existing datasets for sign language translation (§2.2), and provides a comprehensive overview of the state-of-the-art methods employed for automatic translation of sign language videos into text (§2.3).

### 2.1  Use of Bible corpora in NLP

Previous studies have acknowledged the Bible as a valuable resource for language exploration and processing (Mayer and Cysouw, 2014) with good linguistic breadth and depth (Resnik et al., 1999). In machine translation, Bible translations have proven to be a good starting point for machine translation research of many spoken languages, even if eventually one must move to other more useful domains (Liu et al., 2021). In effect, Bible translations have shown their usefulness across different modalities including text (McCarthy et al., 2020) and audio (Black, 2019; Pratap et al., 2023), and for many low resource spoken languages especially in Africa (Dossou and Emezue, 2020; Adelani et al., 2022; Meyer et al., 2022).

Mayer and Cysouw (2014) showcased a corpus of 847 Bibles and McCarthy et al. (2020) increased it significantly both in terms of the number translations (4,000 unique Bible translations) and number of languages (from 1,169 languages to 1,600 languages) it supported thus forming the Johns Hopkins University Bible Corpus (JHUBC). In the audio domain, the CMU Wilderness Speech Dataset (Black, 2019) is a notable resource derived from the New Testaments available on the www.bible.is website. This dataset provides aligned sentence-length text and audio from around 699 different languages. In a similar effort, Meyer et al. (2022) formed BibleTTS: a speech corpus on high-quality Bible translations of 10 African languages. Pratap et al. (2023) expanded both these works and formed the MMS-lab dataset containing Bible translations in 1,107 languages.

### 2.2  Sign Language Translation Datasets

Previous studies on sign language translation were predominantly relying on the RWTH-Phoenix 2014T dataset (Camgoz et al., 2018), which contains 11 hours of weather broadcast footage from the German TV station PHOENIX, covering recordings from 2009 to 2013 (Camgoz et al., 2018; Yin and Read, 2020; De Coster et al., 2021; Zhou et al., 2021; Voskou et al., 2021; Chen et al., 2022b). However, Müller et al. (2023) called into question the scientific value of this dataset. In recent times, TV broadcast datasets have been introduced for several sign languages, including SWISSTXT and VRT (Camgöz et al., 2021), and the BBC-Oxford British Sign Language (BOBSL) dataset (Albanie et al., 2021) for Swiss-German Sign Language, Flemish Sign Language and British Sign Language respectively. Other examples are the How2Sign dataset (Duarte et al., 2021), OpenASL (Shi et al., 2022a) and YouTubeASL (Uthus et al., 2023), featuring American Sign Language.

All datasets mentioned above are bilingual i.e. they contain one single sign language, paired to one spoken language. However, some multilingual datasets have emerged very recently as SP-10 (Yin et al., 2022) and AfriSign (Gueuwou et al., 2023). SP-10 features 10 sign languages but sentences here are extremely short in general (e.g "How are you ?"). AfriSign comprises 6 sign languages which are actually a subset of JWSign. In contrast, JWSign is a valuable resource that surpasses most other sign language translation datasets in terms of duration, signers diversity, and coverage over different sign languages as highlighted in Table 1. Thus, we aim for JWSign to serve as a foundational resource to make sign language translation research more diverse and inclusive going forward.

### 2.3  Sign Language Translation Methods

SLT is an emerging field which aims to translate sign language videos to text/speech and/or vice-versa. One of the main challenges in SLT is finding an efficient and high-quality representation for sign language. This has resulted in many translation architectures using tokenization methods such as human keypoint estimation (Ko et al., 2019; Kim et al., 2020), CNN feature extraction (Zhou et al., 2021; De Coster et al., 2021; Voskou et al., 2021), linguistic glosses (Müller et al., 2023) or phonetic systems such as SignWriting (Jiang et al., 2023). However, most of these are frame-level tokenization methods and assume implicitly that sign language utterances can be considered as sequences of lexical units, while in reality signing uses complex structures in time and 3-dimensional space.

All things considered, 3D CNN window-level

| Dataset | #SL(s) | Vocab | #Hours | Avg | #Signers | Source |
|---|---|---|---|---|---|---|
| PHOENIX (Camgoz et al., 2018) | 1 | 3K | 11 | 4.5 | 9 | TV |
| KETI (Ko et al., 2019) | 1 | 419 | 28 | 6.9 | 14 | lab |
| CSL-Daily (Zhou et al., 2021) | 1 | 2K | 23 | 4.0 | 10 | lab |
| BOBSL (Albanie et al., 2021) | 1 | 78K | 1467 | 4.4 | 39 | TV |
| How2Sign (Duarte et al., 2021) | 1 | 16K | 80 | 8.2 | 11 | lab |
| OpenASL (Shi et al., 2022a) | 1 | 33K | 280 | 10.5 | ≈220 | web |
| YouTubeASL (Uthus et al., 2023) | 1 | 60K | 984 | 4.8 | >2519 | web |
| SP-10 (Yin et al., 2022) | 10 | 17K | 14 | 4.3 | 79 | web |
| AfriSign (Gueuwou et al., 2023) | 6 | 20K | 152 | 18.3 | 160 | web |
| JWSign | 98 | 729K | 2530 | 19.3 | ≈1500 | web |

Table 1: Comparing the JWSign dataset to other common datasets in SLT research. Vocab = Vocabulary of target spoken language i.e. number of unique spoken words, PHOENIX = RWTH Phoenix-2014T, #SL = number of sign language pair(s), #Hours = Total duration of the dataset in hours, Avg = Average duration of a sample in the dataset in seconds.

feature extractors have been reported to reach the best results in this task (Chen et al., 2022a; Müller et al., 2022a; Tarrés et al., 2023). The main component of this approach is an inflated 3D convolutional neural network (I3D) (Carreira and Zisserman, 2017) or S3D (Wei et al., 2016). Originally designed for action recognition (Kay et al., 2017), some works (Varol et al., 2021; Duarte et al., 2022; Chen et al., 2022a) have adapted and fine-tuned these networks for sign language recognition datasets such as BSL-1K (Albanie et al., 2020) and WLASL (Li et al., 2020).

## 3 JWSign

In this section, we give an overview of JWSign (§3.1) and list key statistics, comparing JWSign to other recent datasets (§3.2). Finally, we explain our process of creating fixed data splits for (multilingual) machine translation experiments (§3.3).

### 3.1 Overview

JWSign is made up of verse-aligned Bible translations in 98 sign languages from the Jehovah's Witnesses (JW) website[2]. This wide coverage also extends to the racial identities of the signers, with representation from American Indians/Alaska Natives, Asians, Blacks/African Americans, Hispanics/Latinos, Native Hawaiians/Other Pacific Islanders, and Whites *(in alphabetic order)* (illustrated in in Figure 1). Therefore, we believe that JWSign captures a broad range of signer demo-

[2]https://www.jw.org/

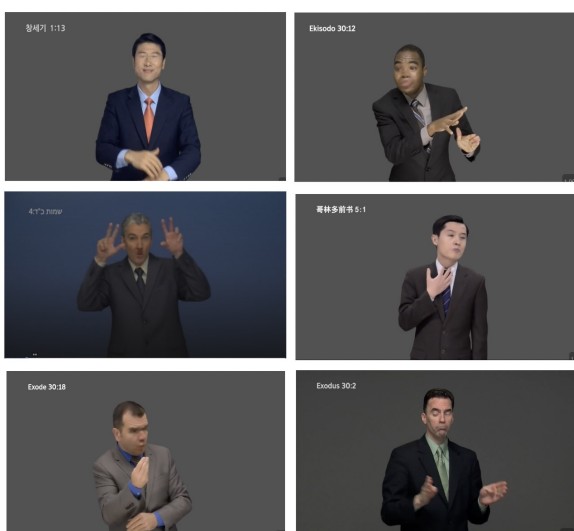

Figure 1: Anecdotal signer diversity in JWSign.

graphics, making it a unique and valuable resource for researchers and practitioners alike.

Translators are either deaf themselves or have grown up in deaf communities, and the recordings are made in a studio on-the-ground in each country. Translations are not only out of English, different spoken languages are used as the source material, depending on the country. Details about the translation process at JW are included in Appendix A.

### 3.2 Statistics of JWSign

JWSign features 98 sign languages spread across all the 7 continents of the world (Table 2). All languages taken together, it has a total duration of 2,530 hours and contains roughly 1,500 individual

| | |
|---|---|
| Europe | 31 |
| Asia | 21 |
| Africa | 19 |
| North America | 11 |
| South America | 11 |
| Oceania | 4 |
| Australia | 1 |

Table 2: Number of sign languages in JWSign per continent.

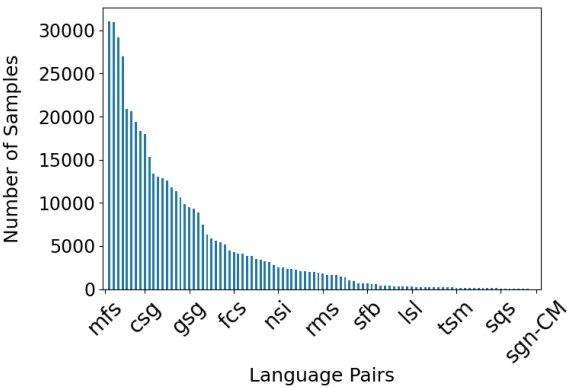

Figure 2: Number of samples per language pair. The x-axis shows language pairs referred to by the ISO code for the sign languages only. All ISO codes are listed in Appendix E. While there are 98 language pairs in total, we show a representative sample of 11, ranging from high-resource to low-resource.

people, according to our automatic analysis.

**Comparison to similar datasets** We show a comparison of JWSign to other datasets in Table 1. JWSign contains more sign languages, covering more geographic regions, than any other dataset we are aware of. For instance, SP-10 (Yin et al., 2022) features 10 sign languages mostly from Europe, and AfriSign (Gueuwou et al., 2023) has 6 sign languages from Africa. JWSign has higher signer diversity (§3.1) than most other datasets. We also observe that samples in other datasets generally are shorter than the average duration in JWSign.

On the other hand, we emphasize that JWSign is a corpus of Bible translations only, hence covering a limited linguistic domain. Other datasets such as BOBSL and YouTubeASL are far more broad, covering many domains and genres. Similarly, when comparing the amount of data available for an individual language pair, JWSign does not always offer the most data. For certain high-resource language pairs, other datasets are considerably larger. For example, BOBSL and YouTubeASL contain ≈1,500 hours and ≈1,000 hours of content in English and British Sign language and American Sign language respectively. Nevertheless, for many language pairs, JWSign is an unparalleled resource for training and evaluating sign language translation models.

**Per-language statistics** JWSign contains at least 2,000 samples for 47 language pairs. The distribution of samples per language pair indicates that some languages are represented better than others (Figure 2). A similar trend is observed with the total duration per language pair (Appendix B).

Naturally, sign languages present a variation in average sample duration across different sign languages, as depicted in Figure 3. This observation sheds light on the linguistic "verbosity" of sign languages, where the same sentence may be signed

in varying lengths across different sign languages. We envision that JWSign enables linguistic studies such as these across many sign languages.

**Cross-lingual frequency** To measure the extent of sample overlaps across different sign languages, we measure how many times each sample (Bible verse) appears across all sign languages. The distribution of this analysis is illustrated in Figure 4.

**Number of individuals** To determine the number of individuals in the dataset, we adopt the signer clustering approach proposed by Pal et al. (2023). We utilize the face recognition toolbox[3] to obtain a 128-dimensional embedding for the signer in each video sample. Then, we use the Density Based Spatial Clustering of Application with Noise (DBSCAN) algorithm (Ester et al., 1996) with $\epsilon = 0.2$ to cluster all embeddings of each sign language. This clustering method is based on the reasonable assumption that no signer can appear in videos for two different sign languages, given that videos are recorded on-the-ground in each country. This yielded a grand total of 1,460 signers[4].

### 3.3 Data splits and automated loader

For each language pair in JWSign we provide a fixed, reproducible split into training, development and test data, tailored towards machine translation as the main use case.

---

[3] https://github.com/ageitgey/face_recognition

[4] We realised this clustering approach works non-optimally for Black/African American and Asian signers and much better on other races. So we expect the ground-truth number of signers in JWSign to be above this value.

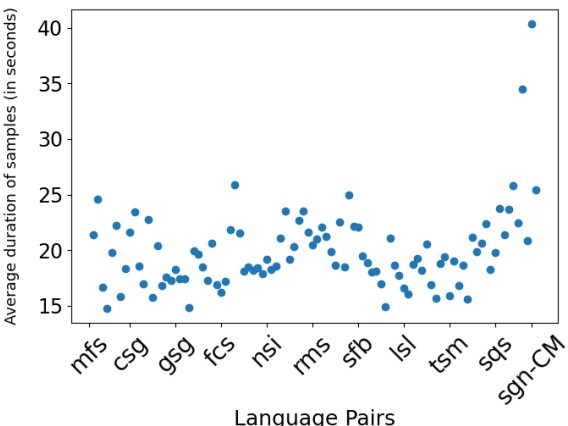

Figure 3: Average duration (in seconds) per language pair. It is worth noting that the two outliers sign languages that exhibit significant deviations from the norm were observed to be those with a very small sample size (less than 10) and long sentences, and are therefore not sufficiently representative of those sign languages.

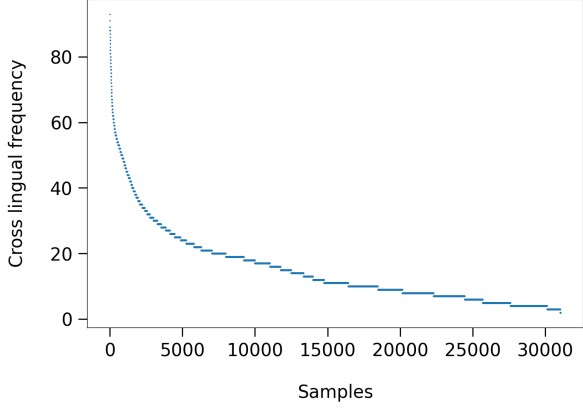

Figure 4: Cross-lingual frequency of Bible verses in JWSign. The y-axis shows the number of sign languages each verse is translated to.

**Splitting procedure** Our method for splitting the data into training, development and test sets is designed to eliminate multilingual "cross-contamination" (the same sentence in two different languages appearing both in the train and test set) as much as possible. Multi-way parallel corpora such as the IWSLT 2017 multilingual task data (Cettolo et al., 2017) (where cross-contamination does exist) are known to paint an overly optimistic picture about the translation quality that can realistically be obtained. A second goal is to maintain a reasonable test set size for machine translation.

We select development and test data based on an analysis of cross-lingual frequency (Figure 4). We minimize the chances of a sample in the test set in one sign language being found in the train set in another language, which could lead to cross-contamination when training a multilingual model and possibly inflate the test set evaluation scores. More details on the splitting procedure are given in Appendix C.

**Automated loader** We do not create new videos nor upload and store them.[5] Instead, JWSign consists of links to Bible verses on the JW website[6] itself and we support it with an automated loader integrated in the Sign Language Datasets library (Moryossef and Müller, 2021) for better accessibility and reproducibilty. More information about the

creation of this automated loader[7] can be found in Appendix D.

## 4 Experimental setup

We perform preliminary machine translation experiments on the JWSign dataset. In this section we explain our preprocessing steps (§4.1), how different models are trained (§4.2) and our method of automatic evaluation (§4.3).

### 4.1 Preprocessing

**Sign language (video) data** All videos have a resolution of 1280 × 720 and frame rate of 29.97 fps. We first resize the videos to a smaller resolution of 256 × 256 pixels and then we apply a center crop to the dimensions of 224 × 224 pixels (the input size expected by the subsequent step). Lastly, we apply color normalization.

The preprocessed videos are then fed into a pretrained I3D model for feature extraction. We use a window size of 64 and a temporal stride of 8, and the particular I3D model we use was fine-tuned by Varol et al. (2021) on an expanded version of the BSL-1K dataset (Albanie et al., 2020), encompassing over 5,000 sign classes.

Using a fine-tuned I3D model, or more generally, a vision-based approach, for feature extraction is motivated by earlier findings. For example, Müller et al. (2022a) and Tarrés et al. (2023) point out that vision-based approaches outperform alternatives such as feature extraction with pose estimation.

What is more, Shi et al. (2022b) have shown

---

[5]Hosting Jehovah's Witnesses data publicly is not allowed: https://www.jw.org/en/terms-of-use/

[6]https://www.jw.org/en/online-help/jw-library-sign-language/

[7]https://github.com/sign-language-processing/datasets/tree/master/sign_language_datasets/datasets/jw_sign

that a pretrained I3D model fine-tuned on a sign language corpus with a greater diversity of signing categories yields more substantial benefits for sign language translation compared to a corpus with fewer signs.

We extract embeddings before the final classification layer, specifically the "mixed_5c" layer. These embeddings are 1024-dimensional vectors that are stacked together, forming a $w \times 1024$-dimensional vector for each sample video, where $w$ is the total number of windows in a sample video.

Finally, for multilingual systems only, a 1024-dimensional vector representing the target spoken language is further appended to the extracted embedding stack. This particular vector serves as a continuous analogue of a tag to indicate the associated target spoken language (Johnson et al., 2017) and it is unique for every spoken language.

**Spoken language (text) data**  We remove special noisy characters as " ∗ " and " + ". The resulting preprocessed text is then tokenized using a Sentencepiece model (Kudo and Richardson, 2018).

## 4.2 Types of models that are compared

In this paper we work exclusively on signed-to-spoken translation, translating from a sign language to a spoken language in all cases. Our models are Transformers (Vaswani et al., 2017) with 6 encoder and 3 decoder layers. Our code is based on Fairseq Sign-to-Text (Tarrés et al., 2023) and is publicly available[8]. All experiments were conducted on a single NVIDIA-A100 GPU.

**Bilingual ("B" systems)**  We developed 36 bilingual models (referred to as **B36**), each focusing on a specific language pair i.e. sign language to spoken language. These language pairs were carefully chosen based on having a substantial number of samples (greater than 1,000 samples) in their respective training sets. We trained the models with a batch size of 32, a learning rate of 1e-3 and applied a dropout rate of 0.3.

We set the Sentencepiece vocabulary size to 1,000 for most language pairs, except for those with a limited number of samples (less than 10,000 in total), where we use a vocabulary size of 500. For languages with a very wide range of characters, such as Chinese and Japanese, we observed many characters are appearing only once (hapax legomena). To counteract this we reduced the character

---

[8] https://github.com/ShesterG/
JWSign-Machine-Translation

coverage to 0.995 and expanded the vocabulary size to 1,500 to accommodate the larger character set. Training was done for a maximum of 100,000 updates.

**Multilingual systems ("M" systems)**  We explore three different multilingual settings. First, we train a single multilingual model using the 36 highest-resource language pairs (same as for B36 above) (**M36**). This enables us to compare the effect of training various language pairs separately and jointly. To optimize the training of multilingual models, we employ a larger batch size of 128 and slightly increase the initial learning rate to 1e-06.

We then attempt a naive model trained on all language pairs in JWSign that have training samples (**M91**). This amounts to 91 different language pairs, excluding seven language pairs in the *Zero* category which do not have any training data. We use these zero-resource language pairs only for testing.

Since we anticipate that the naive multilingual strategy of M91 leads to low translation quality for the low-resource language pairs, we further explore a fine-tuning strategy. For this system (**MFT**), we fine-tune the M36 model jointly on all lower-resource language pairs with training data (i.e. all training data that M36 was *not* trained on, 55 language pairs in total). All hyperparameters are kept the same except that we reset the optimizer accumulator and restart from the 0-th step. This allows us to examine cross-lingual transfer from higher-resource to lower-resource languages. By training on a diverse range of language pairs, we can assess the model's ability to generalize and adapt to unseen sign languages having very little data.

**Clustered families ("C" systems)**  Finally, as another attempt at improving over naive multilingual training, we leverage the phylogeny of spoken languages and sign languages. We cluster the language pairs based on source sign language families (Power et al., 2020; Eberhard et al., 2023) and train on each cluster separately (**CSIG**). Similarly, we cluster the language pairs according to the target spoken language families (Fan et al., 2021) and train on each cluster separately as well (**CSPO**). A sample of each cluster can be found in Table 3 and the full list of clusters is given in Table 6 and Table 7 in the Appendix. The intuition for these clustering experiments is to invoke positive transfer effects stemming from similarities between languages.

| Group | Spoken Languages |
|---|---|
| Germanic | Danish (da), Dutch (nl), English (en), German (de), Norwegian (no), Swedish (sv) |

| Group | Sign Languages |
|---|---|
| Old French | Argentinean (aed), Austrian (asq), Belgian French (sfb), Dutch (dse), Flemish (vgt), French (fsl), German (gsg), Greek (gss), Irish (isg), Israeli (isr), Italian (ise), Mexican (mfs), Quebec (fcs), Spanish (ssp), Swiss German (sgg), Venezuelan (vsl) |

Table 3: Examples for clustering into language families, showing a sign language cluster and a spoken language cluster.

## 4.3 Evaluation

During training, we evaluate models every 2 epochs and select the checkpoint with the highest BLEU score (Papineni et al., 2002) computed with Sacre-BLEU (Post, 2018),[9] aggregated across languages for multilingual models. At test time, using a beam search of size 5, we evaluate all models on the deto-kenized text using BLEU computed with Sacre-BLEU, BLEURT-20 (Pu et al., 2021), and chrF (Popović, 2015). We note that many recent neural metrics, such as COMET (Rei et al., 2020), are not applicable in our case because the source languages (sign languages) are not supported.

## 5 Results and Discussion

Although all language pairs in this work are considered very low-resourced when compared to spoken languages (Goyal et al., 2022), going forward we use the term *High* to refer to language pairs with more than 10,000 training samples, *Medium* for language pairs with training samples between 1,000 and 9,999 inclusive, *Low* for language pairs with training samples between 500 and 999 inclusive, *Very Low* for language pairs with training samples less than 500 and *Zero* for language pairs with no training samples.

Our main findings are summarized in Table 4. Due to limited computational resources, we conducted single runs for all reported results. To give a better overview over our individual results, for some systems on 98 different test sets, we show results aggregated into different training data sizes. For a comprehensive understanding, we have

[9]BLEU+c.mixed+#.1+s.exp+tok.13a+v.1.4.1.

also provided detailed non-aggregate results in Appendix G.

**Performance Variation Across Language Pairs**
The table categorizes the language pairs into different groups based on resource availability, namely *High*, *Medium*, *Low*, *Very Low*, and *Zero*. By examining the performance metrics (BLEU, BLEURT, chrF) within each group, we can observe trends in model performance. For example, the *High* and *Medium* groups tend to have higher scores compared to the *Low*, *Very Low*, and *Zero* groups. This suggests that having a larger training dataset, as indicated by the resource availability, positively impacts the translation quality.

Going into more individual bilingual pair results (Table 8 in Appendix G), the highest BLEU was obtained by Japanese Sign Language to Japanese text (7.08), American Sign Language to English text (4.16) and Chinese Sign Language to Chinese (3.96). This suggests some language pairs are easier for a model to learn, for instance because the grammar of Japanese sign language may be more aligned with spoken Japanese, compared to other language pairs.

**Impact of multilingual training** Here we compare the performance of the "B36" model (bilingual training on 36 language pairs separately) and the "M36" model (multilingual training of one model on the same 36 language pairs together). We observe that our evaluation metrics show conflicting trends, since multilingual training generally reduces BLEU and chrF scores, but increases BLEURT scores. Based on evidence presented in Kocmi et al. (2021) and Freitag et al. (2022), we adopt the view that the neural metric BLEURT is more trustworthy than BLEU and chrF, in the sense of having higher agreement with human judgement. With this interpretation in mind, our results suggest that training on multiple languages simultaneously increases translation quality.

When *Low*, *Very Low* and *Zero* resource language pairs are added to the multilingual training (M91), there is a light drop in scores for *High* and *Medium* language pairs when compared to when they were solely trained together (M36). Thus, while this method may offer advantages for low-resource languages by leveraging knowledge from language pairs with much more data resulting in positive transfer, there is a trade-off between transfer and interference, as increasing the number of

| Models | High | | | Medium | | | Low | | | Very Low | | | Zero | | | Average | | |
|---|---|---|---|---|---|---|---|---|---|---|---|---|---|---|---|---|---|---|
| | BLEU | BLEURT | chrF | BLEU | BLEURT | chrF | BLEU | BLEURT | chrF | BLEU | BLEURT | chrF | BLEU | BLEURT | chrF | BLEU | BLEURT | chrF |
| B36 | 2.37 | 23.36 | 15.87 | 1.65 | 23.43 | 16.07 | - | - | - | - | - | - | - | - | - | 1.89 | 23.4 | 16 |
| M36 | 1.6 | 26.91 | 14.07 | 1.38 | 26.65 | 13.02 | - | - | - | - | - | - | - | - | - | 1.45 | 26.73 | 13.37 |
| M91 | 1.59 | 26.58 | 13.76 | 1.37 | 26.24 | 12.83 | 1.01 | 29.79 | 13.21 | 1 | 27.24 | 12.77 | 0.63 | 30.37 | 9.84 | 1.14 | 27.32 | 12.73 |
| MFT | 0.53 | 16.18 | 13.19 | 0.61 | 22.76 | 13.41 | 1.37 | 22.83 | 16.96 | 1.48 | 24.2 | 15.84 | 1.18 | 22.12 | 14.16 | 1.12 | 22.62 | 14.88 |
| CSIG | 2.37 | 26.04 | 15.35 | 1.82 | 27.28 | 14.98 | 1 | 22.82 | 12.88 | 0.41 | 20.47 | 8.25 | 0.45 | 20.49 | 7.24 | 1.04 | 23.01 | 11.04 |
| CSPO | 2.01 | 27.13 | 14.69 | 1.88 | 28.13 | 15.32 | 1.18 | 24.7 | 12.84 | 0.61 | 21.7 | 10.29 | 0.91 | 22.75 | 11.27 | 1.16 | 24.26 | 12.34 |

Table 4: Evaluation results. B36 = Bilingual Training on 36 language pairs separately, M36 = Multilingual Training on the same 36 language pairs as B36 but jointly, MFT = Fine-tuning of the M36 models on all the remaining 55 language pairs available with available training data, M91 = Joint multilingual training on all the 91 language pairs that have any training data, CSIG = Results of the clustered multilingual models when the source sign languages are from the same group, CSPO = Results of the clustered multilingual models when the target spoken languages are from the same group.

languages in the training set can lead to a decline in performance for the *High* and *Medium* resource language pairs (Arivazhagan et al., 2019).

**Fine-tuning on additional language pairs** The "MFT" model represents the fine-tuning of the "M36" model on all the remaining 55 language pairs with training data available. Comparing its performance with "M36" and "M91", we observe a marked drop in BLEURT scores across all resource categories. On average, "M91" leads to a BLEURT score of 27.32, "MFT" achieves 22.62 on average. This suggests that for incorporating new additional language pairs with limited resources, training these languages from scratch mixed with *High* and *Medium* language pairs is better than a fine-tuning approach.

**Clustered multilingual models** The "CSIG" and "CSPO" models represent the results of clustered multilingual models. In the "CSIG" model, the source sign languages are from the same group, while in the "CSPO" model, the target spoken languages are from the same group. In higher-resource settings, clustered multilingual models perform better than non-clustered multilingual models (M36, MFT, M91). For *High* and *Medium* resource language pairs, CSPO leads to the best BLEURT scores among all model types. For lower-resource settings, the opposite is true and clustering languages based on linguistic philogeny hurts translation quality as measured by BLEURT.

# 6 Conclusion and Future Work

In this study, we introduced JWSign as a unique resource aimed at promoting diversity in sign language processing research which has been so far dominated by few sign languages. We conducted a series of baseline experiments using JWSign to attempt improving the scores of automatic sign translation systems in different scenarios. We demonstrate that multilingual training leads to better translation quality compared to bilingual baselines. On the other hand, our experiments did not show a clear benefit for a fine-tuning approach in lower-resource scenarios. Similarly, we found that clustering data by language family, even though intuitively promising, is only beneficial in higher-resource settings.

More generally, the overall translation quality is still very low. This is in line with other recent studies such as Müller et al. (2022a) who report BLEU scores in a similar range. Regardless, we firmly believe that as we strive to improve translation systems, it is crucial to ensure early diversification of the sign languages used to train these systems. By incorporating a wide range of sign languages during the training phase, we can enhance the inclusivity and effectiveness of the resulting translation systems.

As part of our future research, we aim to develop enhanced models utilizing JWSign that can effectively handle multiple sign languages. Furthermore, JWSign presents a distinctive opportunity to address the existing gaps in sign language processing, such as the development of a sign language identification tool. JWSign can also serve as a valuable tool for linguists to explore and compare various sign languages in an attempt to gain more insights, such as further inquiries into the typological relatedness of sign languages.

## Limitations

There are several limitations to this study that need to be considered.

**Dataset size**  Although JWSign is one of the largest dataset that was designed for Sign Language Translation, it is still quite low-resourced when compared to data in other modalities such as text and/or speech.

**Limited domain**  One limitation of the JWSign dataset is that it is focused on the domain of biblical texts, which may not be representative of other types of sign language communication. This could limit the applicability of the dataset to certain types of sign language translation tasks.

**Translationese effects**  Another limitation of the dataset is the presence of translationese effects, which can occur when translated text or speech sounds unnatural or stilted compared to the original (Barrault et al., 2019). This can be a challenge for sign language translation systems, which must accurately convey the meaning of the source sign language in this case while also producing natural and fluent spoken language.

**Recording conditions**  On top, the videos in the JWSign dataset were recorded in a studio setting, which may not fully capture the complexity and variability of sign language communication in real-world settings. Factors such as lighting, camera angles, and the absence of background noise or visual distractions could affect the sign language production and recognition process in ways that differ from natural communication contexts. This could limit the generalizability of the dataset to real-world sign language translation scenarios.

**Reproducibility**  The dataset is not hosted although we circumvent this with an automated loader to increase accessibility. As long as the original videos and website remain online with stable links, our dataset can be reproduced exactly.

**Uni-directional models**  In this work, we reported baseline scores only for signed-to-spoken translation. We did not experiment at all with translation systems that generate sign language utterances, which is also an important research problem.

## Ethics Statement

**Licensing**  We do not in any way claim ownership of the JW data. We do not create new videos, nor upload or store them. The data strictly and entirely belongs to JW. Instead, we provide links to Bible verses on the JW website itself and we support this with an automated loader to increase accessibility. To the best of our knowledge, we believe that this usage is in accordance with the JW.org terms of use [10], which explicitly allow the distribution of links, as well as downloads/usage of media for "personal and noncommercial purposes". As we neither upload nor copy the actual data, and our aim is to enable researchers to do noncommercial research, we believe these terms are satisfied.

Nevertheless, we have also taken the step of requesting explicit permission by contacting JW's legal branches in the USA and Switzerland (the Office of the General Counsel in New York and the Rechtsabteilung[11] in Thun, Switzerland). However, we have not yet received a reply at the time of writing. Should this permission be refused we certainly plan to abide by their wishes.

Our automated dataset loader includes a usage notice that explicitly informs users of JW's licensing terms.

**Privacy and consent**  We did not reach out to all individuals depicted in our dataset (an estimated 1,500 people) to ask for their consent. We believe our research poses no risk to their privacy because (1) we do not distribute videos and (2) we only train models for signed-to-spoken translation. This means that it is impossible to recover personal information such as faces from a trained model (which we do not share in the first place).

**Algorithmic bias**  On a different note, even though JWSign has signers from all races, the dataset might suffer from other biases such as gender, age representation and handedness. Models trained here are far from usable and reliable, and thus cannot replace a human sign language interpreter.

---

[10] https://www.jw.org/en/terms-use/
[11] https://www.jw.org/de/rechtlich/rechtsabteilungen-kontakt/schweiz/

## Acknowledgements

We thank Antonis Anastasopoulos and Chris Emezue for feedback on the manuscript. Also, we would like to thank Google Cloud for providing us access to computational resources through free cloud credits. MM has received funding from the EU Horizon 2020 project EASIER (grant agreement number 101016982).

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

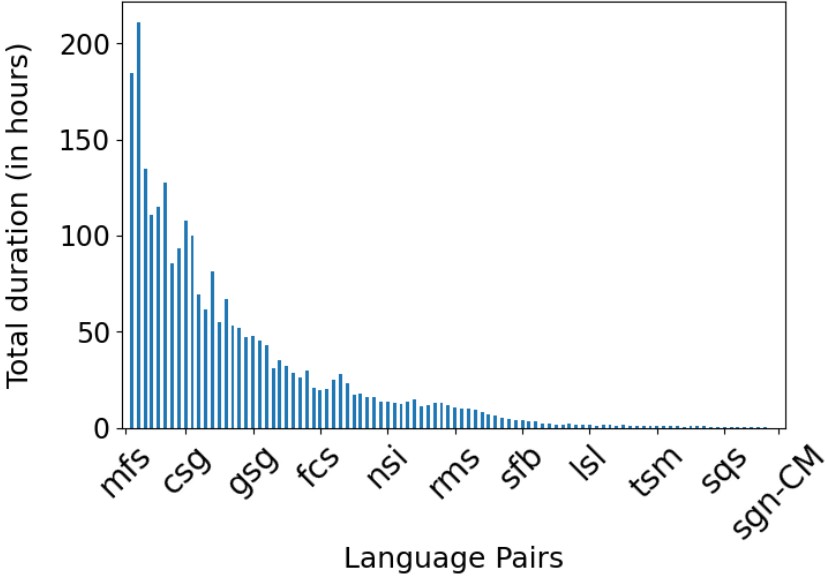

Figure 5: Total duration (in hours) per language pair.

## A Translation process at Jehovah's Witnesses

The Witnesses' approach to sign language translation is thorough and collaborative[12]. Newly recruited translators receive extensive training in translation principles and work in teams, where each member performs a specific role such as translating, checking, or proofreading the material. To ensure the highest quality of translation, a panel of deaf individuals from diverse backgrounds and locations review the translation and provide valuable feedback to refine the signs and expressions used in the final video. This step guarantees that the message is conveyed accurately and naturally.

In addition to their translation work, the sign-language translators participate in congregation meetings and hold Bible studies with non-Jehovah's Witnesses members of the deaf community, enabling them to stay abreast of language developments and improve their skills. This diligent approach to sign-language translation ensures translators stay up-to-date with the language.

The videos are recorded in a studio with proper lighting and the translation is done from the region's official spoken language to the country's sign language verse by verse. This work is incremental and still ongoing - as of January 2023, the complete Bible is only available in three sign languages (American Sign Language, Brazilian Sign Language and Mexican Sign Language).

## B Additional Statistics

The distribution of the total number of hours in each language pair can be found at Figure 5.

## C Details of data splitting procedure

First, we sort the samples by cross-lingual frequency in descending order, based on the number of sign languages in which they appear. This ensures that samples with the most overlap across sign languages will be found at the top of the list, while samples with the least overlap will be found at the bottom (Figure 4). We proceed by partitioning the samples into three distinct and non-overlapping buckets. The test bucket consists of the 1,500 most frequently occurring samples, followed by the dev bucket containing the next 1,500 samples, and finally the train bucket comprising the remaining samples. For each language, the test, dev, and train sets are formed by intersecting the language-specific samples with their corresponding buckets. This method helps to eliminate the possibility of a sample in the test set in one sign language being found in the train set in another language, which could lead to cross-contamination when training a multilingual model and possibly inflate the test set evaluation scores.

---

[12]https://www.jw.org/en/jehovahs-witnesses/activities/publishing/sign-language-translation/

## D   Development of automated loader

To create this loader, we followed a few key steps. First, we created an index file that contains a comprehensive list of verses and their attributes, such as the video URL on the JW website, start and end times of the verse in the video, and a link to the corresponding written text on the JW website. We ensured that all selected videos had a frame rate of 29.970 fps (the most common fps used in the videos on the website) and resolution of 1280x720, and we eliminated any duplicates. The index file was stored in JSON format, which has the advantage of being easily updatable when the website gets updated.

Next, we developed a script that utilizes the information in the index file to automatically load frames/poses and corresponding text, aligning them appropriately to form a dataset. With this loader, users have the option to form a dataset for a specific sign language, a set of sign languages, or all sign languages as needed. The human poses estimation can be obtained from Mediapipe Holistic (Lugaresi et al., 2019) or OpenPose (Cao et al., 2017). Human pose estimation refers to a computer vision task that involves detecting, predicting, and monitoring the positions of various joints and body parts. Both OpenPose and Mediapipe Holistic are capable of detecting various keypoints present in videos, including those on the face, hands, and body.

We believe that the automated loader for JWSign integrated in the Sign Language Datasets library will streamline the process of accessing sign language data.

## E   Dataset Statistics

Table 5 highlights the statistics about all the 98 sign languages in JWSign.

## F   Language Groupings

Table 7 and Table 6 highlights the sign language groups and spoken language groups respectively, used during Clustering Families Training.

## G   Results detailed

Table 8 shows the evaluation results on model B36 and model M36, while Table 9 shows the evaluation results on model M91, CSIG, CSPO and MFT.

| sign language | iso | spoken | samples | duration | avg | train / dev / test |
|---|---|---|---|---|---|---|
| Mexican | mfs | es | 31056 | 184.473 | 22 | 28057 / 1500 / 1499 |
| Brazilian | bzs | pt-br | 30949 | 211.135 | 25 | 27957 / 1494 / 1498 |
| American | ase | en | 29150 | 134.655 | 17 | 26358 / 1340 / 1449 |
| Russian | rsl | ru | 26949 | 110.571 | 15 | 24109 / 1449 / 1391 |
| Italian | ise | it | 20882 | 114.923 | 20 | 19376 / 796 / 710 |
| Colombian | csn | es | 20644 | 127.63 | 23 | 18506 / 1151 / 987 |
| Spanish | ssp | es | 19394 | 85.207 | 16 | 16416 / 1483 / 1495 |
| Korean | kvk | ko | 18287 | 93.322 | 19 | 16030 / 1104 / 1153 |
| Argentinean | aed | es | 17818 | 106.856 | 22 | 14946 / 1435 / 1437 |
| Chilean | csg | es | 15357 | 100.15 | 24 | 12845 / 1282 / 1230 |
| Ecuadorian | ecs | es | 13331 | 68.873 | 19 | 10577 / 1368 / 1386 |
| Polish | pso | pl | 12994 | 61.294 | 17 | 10085 / 1435 / 1474 |
| Peruvian | prl | es | 12843 | 81.079 | 23 | 9890 / 1456 / 1497 |
| British | bfi | en | 12538 | 54.925 | 16 | 9557 / 1485 / 1496 |
| Japanese | jsl | ja | 11832 | 67.154 | 21 | 8929 / 1409 / 1494 |
| Indian | ins | en | 11384 | 53.298 | 17 | 8609 / 1358 / 1417 |
| Venezuelan | vsl | es | 10634 | 51.821 | 18 | 7720 / 1420 / 1494 |
| South African | sfs | en | 9837 | 47.159 | 18 | 7046 / 1352 / 1439 |
| Zimbabwe | zib | en | 9463 | 47.889 | 19 | 6787 / 1313 / 1363 |
| German | gsg | de | 9335 | 45.164 | 18 | 6412 / 1432 / 1491 |
| Malawi | sgn-MW | ny | 8849 | 42.815 | 18 | 6266 / 1246 / 1337 |
| French | fsl | fr | 7415 | 30.545 | 15 | 4735 / 1257 / 1423 |
| Finnish | fse | fi | 6303 | 34.883 | 20 | 3432 / 1394 / 1477 |
| Angolan | sgn-AO | pt-pt | 5867 | 32.05 | 20 | 3490 / 1100 / 1277 |
| Australian | asf | en | 5597 | 28.75 | 19 | 2900 / 1266 / 1431 |
| Cuban | csf | es | 5406 | 25.968 | 18 | 2868 / 1145 / 1393 |
| Indonesian | inl | id | 5201 | 29.651 | 21 | 3000 / 921 / 1280 |
| Filipino | psp | en | 4406 | 20.638 | 17 | 1928 / 1096 / 1382 |
| Chinese | csl | zh-CN | 4280 | 19.278 | 17 | 2143 / 778 / 1359 |
| Zambian | zsl | en | 4067 | 24.666 | 22 | 1920 / 886 / 1261 |
| Quebec | fcs | fr | 4058 | 19.518 | 18 | 1604 / 1056 / 1398 |
| Bolivian | bvl | es | 3881 | 27.864 | 26 | 1411 / 1117 / 1352 |
| Paraguayan | pys | gn | 3810 | 22.837 | 22 | 1506 / 979 / 1325 |
| Kenyan | xki | en | 3452 | 17.351 | 19 | 1286 / 927 / 1239 |
| Czech | cse | cs | 3412 | 17.537 | 19 | 1498 / 593 / 1321 |
| Ghanaian | gse | en | 3185 | 16.062 | 19 | 1965 / 378 / 842 |
| Hungarian | hsh | hu | 3125 | 16.011 | 19 | 851 / 837 / 1437 |
| Taiwanese | tss | zh-TW | 2754 | 13.707 | 18 | 799 / 722 / 1233 |
| Swedish | swl | sv | 2540 | 13.532 | 20 | 768 / 519 / 1253 |
| Portuguese | psr | pt-pt | 2368 | 12.208 | 19 | 466 / 593 / 1309 |
| Nigerian | nsi | en | 2347 | 11.898 | 19 | 774 / 593 / 980 |
| Slovak | svk | sk | 2302 | 13.48 | 22 | 450 / 505 / 1347 |
| Honduras | hds | es | 2290 | 14.964 | 24 | 594 / 405 / 1291 |
| Costa Rican | csr | es | 2099 | 11.177 | 20 | 478 / 338 / 1283 |
| Guatemalan | gsm | es | 2081 | 11.763 | 21 | 517 / 309 / 1255 |
| Panamanian | lsp | es | 2025 | 12.741 | 23 | 397 / 326 / 1302 |
| Nicaraguan | ncs | es | 2013 | 13.152 | 24 | 534 / 278 / 1201 |
| Madagascar | mzc | mg | 1935 | 11.624 | 22 | 321 / 577 / 1037 |
| Salvadoran | esn | es | 1806 | 10.289 | 21 | 458 / 232 / 1116 |
| Romanian | rms | ro | 1647 | 9.632 | 22 | 126 / 339 / 1182 |

| | | | | | | |
|---|---|---|---|---|---|---|
| Mozambican | mzy | pt-pt | 1628 | 9.596 | 22 | 241 / 311 / 1076 |
| Greek | gss | el | 1627 | 10.011 | 23 | 102 / 339 / 1186 |
| Thai | tsq | th | 1456 | 8.049 | 20 | 143 / 201 / 1112 |
| Congolese | sgn-CD | fr | 1334 | 6.914 | 19 | 245 / 216 / 873 |
| Ivorian | sgn-CI | fr | 977 | 6.122 | 23 | 103 / 151 / 723 |
| Croatian | csq | hr | 960 | 4.922 | 19 | 85 / 113 / 762 |
| Myanmar | sgn-MM | my | 666 | 4.623 | 25 | 91 / 99 / 476 |
| Tanzanian | tza | sw | 651 | 4.007 | 23 | 80 / 104 / 467 |
| Malaysian | xml | ms | 643 | 3.936 | 23 | 57 / 84 / 502 |
| Belgian French | sfb | fr | 609 | 3.292 | 20 | 56 / 67 / 486 |
| Vietnamese | hab | vi | 606 | 3.18 | 19 | 38 / 90 / 478 |
| Uruguayan | ugy | es | 398 | 1.994 | 19 | 19 / 34 / 345 |
| New Zealand | nzs | en | 368 | 1.849 | 19 | 52 / 36 / 280 |
| Irish | isg | en | 362 | 1.708 | 17 | 23 / 36 / 303 |
| Dutch | dse | nl | 330 | 1.366 | 15 | 8 / 27 / 295 |
| Serbian | sgn-RS | sr | 328 | 1.922 | 22 | 62 / 46 / 220 |
| Albanian | sqk | sq | 310 | 1.605 | 19 | 1 / 10 / 299 |
| Norwegian | nsl | no | 275 | 1.354 | 18 | 32 / 29 / 214 |
| Swiss German | sgg | de | 270 | 1.244 | 17 | 12 / 29 / 229 |
| Latvian | lsl | lv | 257 | 1.149 | 17 | 6 / 22 / 229 |
| Austrian | asq | de | 252 | 1.309 | 19 | 1 / 15 / 236 |
| Danish | dsl | da | 243 | 1.301 | 20 | 48 / 32 / 163 |
| Ugandan | ugn | en | 236 | 1.192 | 19 | 10 / 27 / 199 |
| Nepali | nsp | ne | 228 | 1.301 | 21 | 5 / 12 / 211 |
| Jamaican | jls | en | 227 | 1.067 | 17 | 16 / 31 / 180 |
| Flemish | vgt | nl | 183 | 0.986 | 20 | 7 / 14 / 162 |
| Israeli | isr | he | 177 | 0.782 | 16 | 20 / 9 / 148 |
| Turkish | tsm | tr | 173 | 0.913 | 19 | 9 / 6 / 158 |
| Lithuanian | lls | lt | 170 | 0.793 | 17 | 5 / 7 / 158 |
| Ethiopian | eth | am | 162 | 0.84 | 19 | 12 / 18 / 132 |
| Cambodian | sgn-KH | km | 145 | 0.629 | 16 | 1 / 5 / 139 |
| Mongolian | msr | mn | 129 | 0.571 | 16 | 1 / 8 / 120 |
| Armenian | aen | hy | 129 | 0.763 | 22 | 5 / 6 / 118 |
| Estonian | eso | et | 122 | 0.674 | 20 | 6 / 9 / 107 |
| Melanesian | sgn-PG | en | 117 | 0.67 | 21 | 9 / 7 / 101 |
| Slovenian | sgn-SI | sl | 96 | 0.596 | 23 | 1 / 2 / 93 |
| Suriname | sgn-SR | nl | 96 | 0.487 | 19 | 2 / 2 / 92 |
| Bulgarian | bqn | bg | 87 | 0.479 | 20 | 3 / 2 / 82 |
| Rwandan | sgn-RW | rw | 77 | 0.402 | 19 | 13 / 20 / 44 |
| Sri Lankan | sqs | si | 61 | 0.403 | 24 | 1 / 1 / 59 |
| Hong Kong | hks | zh-TW | 60 | 0.356 | 22 | 2 / 1 / 57 |
| Singapore | sls | en | 33 | 0.217 | 24 | 0 / 0 / 33 |
| Fiji | sgn-FJ | en | 30 | 0.215 | 26 | 0 / 0 / 30 |
| Lebanese | sgn-LB | ar | 28 | 0.174 | 23 | 0 / 0 / 28 |
| Samoan | sgn-WS | sm | 8 | 0.077 | 35 | 0 / 0 / 8 |
| Burundi | sgn-BI | fr | 3 | 0.017 | 21 | 0 / 0 / 3 |
| Mauritian | lsy | en | 2 | 0.022 | 41 | 0 / 0 / 2 |
| Cameroon | sgn-CM | fr | 2 | 0.014 | 26 | 0 / 0 / 2 |
| TOTAL | 98 | 51 | 472529 | 2530.262 | 19.3 | 341334 / 52052 / 79143 |

Table 5: Comparing the different sign languages in JWSign. iso = ISO 639-3 sign language code, samples = total number of videos, duration = total duration of all videos (in hours), avg = average duration of samples (in seconds).

| Group | Languages |
|---|---|
| Chinese | Chinese mandarin simplified (zh-CN), Chinese mandarin traditional (zh-TW), Japanese (ja), Korean (ko), Vietnamese (vi) |
| Germanic | Danish (da), Dutch (nl), English (en), German (de), Norwegian (no), Swedish (sv) |
| Malayo-Polynesian | Indonesian (id), Malagasy (mg), Malay (ms) |
| Niger-Congo | Amharic (am), Chichewa (ny), Kinyarwanda (rw), Swahili (sw) |
| Romance | French (fr), Italian (it), Portuguese-brazil (pt-br), Portuguese-portugal (pt-pt), Romanian (ro), Spanish (es) |
| Slavic | Bulgarian (bg), Croatian (hr), Czech (cs), Polish (pl), Russian (ru), Serbian-roman (sr), Slovak (sk), Slovenian (sl) |
| Uralic | Estonian (et), Finnish (fi), Hungarian (hu), Latvian (lv), Lithuanian (lt) |
| Other | Albanian (sq), Arabic (ar), Armenian (hy), Cambodian (km), Greek (el), Guarani-paraguayan (gn), Hebrew (he), Mongolian (mn), Myanmar (my), Nepali (ne), Samoan (sm), Sinhala (si), Thai (th), Turkish (tr) |

Table 6: Spoken languages Groups

| Group | Languages |
|---|---|
| America | American (ase), Bolivian (bvl), Burundi (sgn-BI), Cambodian (sgn-KH), Cameroon (sgn-CM), Colombian (csn), Congolese (sgn-CD), Costa Rican (csr), Ecuadorian (ecs), Ethiopian (eth), Filipino (psp), Ghanaian (gse), Guatemalan (gsm), Indonesian (inl), Ivorian (sgn-CI), Jamaican (jls), Kenyan (xki), Malawi (sgn-MW), Malaysian (xml), Myanmar (sgn-MM), Nigerian (nsi), Panamanian (lsp), Peruvian (prl), Rwandan (sgn-RW), Salvadoran (esn), Singapore (sls), Sri Lankan (sqs), Thai (tsq), Ugandan (ugn), Zambian (zsl), Zimbabwe (zib) |
| British | Australian (asf), British (bfi), Croatian (csq), Fiji (sgn-FJ), Indian (ins), Melanesian (sgn-PG), Nepali (nsp), New Zealand (nzs), Samoan (sgn-WS), Serbian (sgn-RS), Slovenian (sgn-SI), South African (sfs) |
| Chinese | Chinese (csl), Hong Kong (hks), Japanese (jsl), Korean (kvk), Taiwanese (tss) |
| Old French | Argentinean (aed), Austrian (asq), Belgian French (sfb), Dutch (dse), Flemish (vgt), French (fsl), German (gsg), Greek (gss), Irish (isg), Israeli (isr), Italian (ise), Mexican (mfs), Quebec (fcs), Spanish (ssp), Swiss German (sgg), Venezuelan (vsl) |
| Polish | Bulgarian (bqn), Czech (cse), Estonian (eso), Hungarian (hsh), Latvian (lsl), Lithuanian (lls), Mongolian (msr), Polish (pso), Romanian (rms), Russian (rsl), Slovak (svk) |
| Swedish | Brazilian (bzs), Danish (dsl), Finnish (fse), Madagascar (mzc), Norwegian (nsl), Portuguese (psr), Swedish (swl) |
| Uruguay | Chilean (csg), Paraguayan (pys), Uruguayan (ugy) |
| Other | Angolan (sgn-AO), Honduras (hds), Nicaraguan (ncs), Suriname (sgn-SR), Turkish (tsm), Vietnamese (hab), Lebanese (sgn-LB), Albanian (sqk), Armenian (aen), Mauritian (lsy), Mozambican (mzy), Tanzanian (tza), Cuban (csf) |

Table 7: Sign languages Groups

| Language Pair | B36 | | | M36 | | |
|---|---|---|---|---|---|---|
| | BLEU | BLEURT | chrF | BLEU | BLEURT | chrF |
| mfs–>es | 2.33 | 21.18 | 15.33 | 1.62 | 27.75 | 14.74 |
| bzs–>pt-br | 2.12 | 18.7 | 14.96 | 1.47 | 21.76 | 12.75 |
| ase–>en | 4.16 | 37.49 | 18.99 | 2.48 | 36.91 | 15.46 |
| rsl–>ru | 3.09 | 26.38 | 17.81 | 1.14 | 24.74 | 12.21 |
| ise–>it | 2.58 | 25.49 | 16.3 | 1.33 | 27.4 | 12.45 |
| csn–>es | 2.04 | 23.73 | 14.07 | 1.32 | 28.03 | 14.48 |
| ssp–>es | 2.64 | 20.42 | 17.52 | 1.48 | 27.17 | 14.68 |
| kvk–>ko | 1.37 | 32.38 | 9.88 | 3.68 | 31.88 | 16.56 |
| aed–>es | 2.2 | 21.37 | 15.48 | 1.44 | 26.91 | 14.68 |
| csg–>es | 2.38 | 19.1 | 16.63 | 1.36 | 28.22 | 14.8 |
| ecs–>es | 1.76 | 16.94 | 16.38 | 1.52 | 26.36 | 14.64 |
| pso–>pl | 1.78 | 17.15 | 17.06 | 0.38 | 15.76 | 11.44 |
| prl–>es | 1.92 | 19.57 | 15.69 | 1.44 | 27.1 | 14.69 |
| bfi–>en | 2.89 | 36.02 | 18.19 | 1.99 | 36.11 | 15.05 |
| jsl–>ja | 7.08 | 23.96 | 14.77 | 5.44 | 25.49 | 14.83 |
| ins–>en | 2.87 | 35.01 | 18.43 | 1.82 | 36.25 | 15.01 |
| vsl–>es | 1.83 | 15.74 | 16.87 | 1.3 | 26.94 | 14.03 |
| sfs–>en | 1.99 | 33.95 | 16.16 | 1.75 | 36.39 | 14.89 |
| zib–>en | 1.47 | 34.43 | 14.92 | 1.55 | 36.69 | 14.39 |
| gsg–>de | 1.22 | 19.34 | 16.58 | 0.58 | 19.9 | 12.12 |
| sgn-MW–>ny | 0.93 | 26.81 | 19.06 | 0.31 | 23.95 | 12.17 |
| fsl–>fr | 1.36 | 0.48 | 16.28 | 0.47 | -2.24 | 8.42 |
| sgn-AO–>pt-pt | 0.73 | 13.18 | 13.03 | 0.78 | 19.39 | 11.72 |
| fse–>fi | 1.2 | 22.33 | 18.45 | 0.35 | 15.41 | 8.36 |
| inl–>id | 1 | 45.08 | 19.16 | 0.23 | 40.53 | 13.02 |
| asf–>en | 1.52 | 32.72 | 15.5 | 1.75 | 37.18 | 15.57 |
| csf–>es | 1.02 | 12.97 | 14.75 | 1.07 | 26.49 | 14.06 |
| csl–>zh-CN | 3.96 | 33.7 | 14.49 | 4.92 | 34.14 | 16.32 |
| gse–>en | 1.22 | 31.5 | 16.14 | 1.15 | 35.94 | 14.92 |
| psp–>en | 1.11 | 31.88 | 16.97 | 1.68 | 37.1 | 14.87 |
| zsl–>en | 1.08 | 27.51 | 15.71 | 1.22 | 35.58 | 14.41 |
| fcs–>fr | 0.8 | -1 | 14.98 | 0.51 | -1.76 | 8.32 |
| pys–>gn | 0.64 | 17.65 | 15.73 | 0.17 | 13.02 | 12.53 |
| cse–>cs | 0.38 | 11.19 | 12.84 | 0.02 | 16.38 | 4.09 |
| bvl–>es | 0.57 | 9.24 | 14.81 | 1.08 | 27.21 | 14.31 |
| xki–>en | 0.85 | 28.98 | 16.11 | 1.53 | 36.33 | 14.42 |

Table 8: Evaluation Results on B36 and M36.

| Language Pair | M91 | | | CSIG | | | CSPO | | | MFT | | |
|---|---|---|---|---|---|---|---|---|---|---|---|---|
| | BLEU | BLEURT | chrF | BLEU | BLEURT | chrF | BLEU | BLEURT | chrF | BLEU | BLEURT | chrF |
| mfs->es | 2 | 28.46 | 14.42 | 2.91 | 26.79 | 16.02 | 2.04 | 27.47 | 14.56 | 0.74 | 15.73 | 16.08 |
| bzs->pt-br | 1.47 | 20.42 | 13.34 | 3.03 | 20.3 | 17.02 | 1.68 | 22.83 | 13.66 | 0.58 | 10.08 | 16.75 |
| ase->en | 1.84 | 37.03 | 15.09 | 3.06 | 37.86 | 16.52 | 3.86 | 39.08 | 18.91 | 0.87 | 32.06 | 15.35 |
| rsl->ru | 1.21 | 22.58 | 12.63 | 3.12 | 25.89 | 17.8 | 3.73 | 26.88 | 18.63 | 0.04 | 7.6 | 1.48 |
| ise->it | 1.33 | 29.38 | 12.86 | 2.37 | 28.06 | 14.76 | 1.33 | 26.39 | 12.76 | 0.03 | 10.41 | 8.12 |
| csn->es | 1.47 | 27.87 | 13.92 | 2.16 | 26.96 | 14.35 | 1.73 | 26.04 | 14.63 | 0.66 | 15.93 | 16.44 |
| ssp->es | 1.6 | 27.23 | 14.15 | 2.47 | 26.23 | 15.63 | 1.75 | 26.1 | 14.44 | 0.53 | 16.66 | 15.49 |
| kvk->ko | 2.85 | 32.26 | 14.84 | 1.27 | 29.81 | 8.63 | 1.43 | 30.3 | 9.23 | 0.83 | 22.59 | 10.66 |
| aed->es | 1.66 | 27.39 | 14.04 | 2.34 | 27.04 | 15.29 | 1.4 | 26.38 | 14.11 | 0.63 | 15.8 | 16.28 |
| csg->es | 1.67 | 28.11 | 14.65 | 2.31 | 16.54 | 17.73 | 1.55 | 26.67 | 14.52 | 0.66 | 17.35 | 16.81 |
| ecs->es | 1.58 | 26.81 | 14.07 | 1.7 | 26.52 | 14.17 | 1.65 | 25.55 | 14.38 | 0.72 | 17.38 | 16.08 |
| pso->pl | 0.36 | 11.46 | 11.12 | 1.69 | 20.49 | 16.3 | 1.92 | 21.92 | 16.4 | 0.05 | 12.51 | 8.76 |
| prl->es | 1.74 | 27.05 | 14.19 | 2.02 | 27.24 | 14.6 | 1.51 | 25.98 | 14.39 | 0.61 | 17.93 | 16.78 |
| bfi->en | 1.62 | 35.87 | 14.55 | 3.13 | 36.26 | 18.89 | 3.04 | 38.01 | 17.98 | 1.05 | 31.82 | 15.54 |
| jsl->ja | 5.06 | 24.97 | 13.78 | 6.55 | 24.3 | 13.19 | 6.12 | 23.7 | 12.52 | 1.11 | 22.09 | 12.98 |
| ins->en | 1.47 | 35.26 | 14.18 | 3.14 | 36.09 | 18.6 | 2.9 | 37.94 | 17.98 | 0.85 | 32.29 | 15.99 |
| vsl->es | 1.37 | 27.09 | 13.69 | 2.05 | 26.32 | 14.84 | 1.42 | 26.48 | 13.89 | 0.58 | 16.64 | 15.79 |
| sfs->en | 1.45 | 35.24 | 14.24 | 2.32 | 35.54 | 18.15 | 2.64 | 37.23 | 17.63 | 0.88 | 32.06 | 15.74 |
| zib->en | 1.18 | 35.76 | 13.84 | 1.63 | 36.14 | 14.7 | 2.28 | 36.69 | 16.71 | 1.06 | 31.73 | 15.4 |
| gsg->de | 0.37 | 19.85 | 12.79 | 0.91 | 22.51 | 13.99 | 1.12 | 23.48 | 16.19 | 0.09 | 22.76 | 13.07 |
| sgn-MW->ny | 0.21 | 24.53 | 13.88 | 0.48 | 28.04 | 15.3 | 1.31 | 27.46 | 18.27 | 0.03 | 22.75 | 7.31 |
| fsl->fr | 0.69 | 1.69 | 9.23 | 1.1 | 3.89 | 12.2 | 0.74 | 0.85 | 9.78 | 0.47 | -2.69 | 14.1 |
| sgn-AO->pt-pt | 0.86 | 18.1 | 12.54 | 0.88 | 10.7 | 13.32 | 0.89 | 21.31 | 12.47 | 0.68 | 10.42 | 13.98 |
| fse->fi | 0.45 | 22.22 | 7.96 | 1.46 | 29.43 | 16.56 | 1.35 | 27.57 | 18.22 | 0.08 | 19.57 | 10.83 |
| inl->id | 0.2 | 33.68 | 12.86 | 0.62 | 45.08 | 17.18 | 1.59 | 46.86 | 20.04 | 0.03 | 31.09 | 8.51 |
| asf->en | 1.39 | 36.32 | 14.75 | 2.23 | 34.85 | 17.93 | 2.31 | 37.48 | 17.31 | 0.92 | 32.35 | 15.85 |
| csf->es | 1.23 | 25.85 | 13.31 | 0.74 | 13.83 | 13.8 | 1.37 | 24.57 | 13.74 | 0.51 | 16.9 | 15.13 |
| csl->zh-CN | 6.54 | 29.07 | 16.48 | 4.25 | 36.89 | 13.93 | 3.52 | 36.21 | 13.2 | 0.18 | 28.8 | 5.82 |
| gse->en | 1.11 | 35.17 | 14.55 | 1.56 | 35.43 | 15.52 | 1.84 | 36.38 | 16.63 | 0.98 | 31.39 | 15.89 |
| psp->en | 1.43 | 36.93 | 14.2 | 1.91 | 36.65 | 15.13 | 2.09 | 37.35 | 16.41 | 0.93 | 31.85 | 15.71 |
| zsl->en | 1.06 | 35.08 | 14.03 | 1.44 | 35.07 | 15.07 | 1.68 | 34.54 | 16.16 | 0.78 | 30.55 | 14.97 |
| fcs->fr | 0.7 | 1.77 | 9.06 | 1.03 | 4.61 | 11.49 | 0.58 | 0.29 | 9.42 | 0.6 | -1.99 | 13.86 |
| pys->gn | 0.25 | 13.86 | 11.2 | 0.52 | 17.78 | 14.75 | 1.12 | 17.24 | 16.53 | 0.09 | 29.14 | 8.61 |
| cse->cs | 0.04 | 12.83 | 4.94 | 0.47 | 16.1 | 11.82 | 0.62 | 16.02 | 11.62 | 0.12 | 9.82 | 7.99 |
| bvl->es | 1.22 | 27.24 | 13.85 | 1.53 | 26.52 | 13.62 | 1.42 | 25.93 | 14.11 | 0.84 | 17.68 | 16.8 |
| xki->en | 1.17 | 34.31 | 13.83 | 1.68 | 35.51 | 15.02 | 1.64 | 35.56 | 16.39 | 1.15 | 31.32 | 15.27 |
| hsh->hu | 0.02 | 42.61 | 6.11 | 1.23 | 30.39 | 14.11 | 1.25 | 27.09 | 15.33 | 0.95 | 27.62 | 15.56 |
| tss->zh-TW | 2.02 | 29.33 | 26.32 | 0.68 | 17.28 | 6.62 | 0.67 | 18.34 | 6.68 | 4.6 | 23.54 | 21.93 |
| nsi->en | 1.31 | 36.11 | 14.41 | 1.88 | 36.06 | 15.29 | 2.43 | 36.64 | 16.94 | 1.29 | 32.68 | 16.54 |
| swl->sv | 0.02 | 21.11 | 3.93 | 0.75 | 22.69 | 12.88 | 0.25 | 14.93 | 9.05 | 0.58 | 19.78 | 15.12 |
| hds->es | 0.97 | 26.6 | 13.81 | 0.5 | 13.37 | 13.93 | 0.99 | 25.81 | 13.96 | 0.67 | 18.72 | 16.55 |
| ncs->es | 1.25 | 26.32 | 14.04 | 0.61 | 13.79 | 14.08 | 1.18 | 24.91 | 14 | 0.73 | 18.89 | 16.65 |
| gsm->es | 1.49 | 26.48 | 13.88 | 1.35 | 26.15 | 13.26 | 1.5 | 25.21 | 13.91 | 0.76 | 18.6 | 16.36 |
| csr->es | 1.1 | 25.35 | 13.56 | 1.1 | 25.04 | 13.05 | 0.99 | 24.54 | 13.62 | 0.63 | 18.26 | 16.15 |
| psr->pt-pt | 0.95 | 18.77 | 13.09 | 1.16 | 16.82 | 14.36 | 0.95 | 21.1 | 12.28 | 0.69 | 11.12 | 14.85 |
| esn->es | 1.21 | 26.59 | 13.79 | 1.15 | 26.15 | 13.23 | 1.07 | 24.64 | 13.88 | 0.71 | 18.52 | 16.36 |
| svk->sk | 0.04 | 22.33 | 6.29 | 0.38 | 17.47 | 11.66 | 0.26 | 16.92 | 10.65 | 0.92 | 16.33 | 12.96 |
| lsp->es | 1.29 | 26.68 | 14.02 | 1.18 | 25.11 | 13.84 | 1.11 | 24.59 | 14.21 | 0.82 | 18.56 | 17.03 |
| mzc->mg | 0.02 | 29.21 | 9.94 | 0.17 | 28.19 | 14.2 | 0.64 | 28.87 | 18.65 | 0.52 | 28.93 | 19.36 |
| sgn-CD->fr | 0.64 | 1.9 | 8.93 | 0.01 | 4.28 | 0.37 | 0.52 | -0.49 | 9.07 | 0.48 | -2.01 | 14.85 |
| mzy->pt-pt | 0.55 | 18.28 | 12.6 | 0.6 | 10.27 | 13.33 | 0.75 | 22.21 | 12.54 | 0.58 | 10.58 | 14.8 |
| tsq->th | 1.34 | 40.02 | 25.54 | 0.03 | 15.79 | 4.83 | 1.24 | 15.17 | 13.47 | 7.09 | 38.71 | 27.64 |
| rms->ro | 1.19 | 23.57 | 8.74 | 0.06 | 18.17 | 9.11 | 0.01 | 25.6 | 4.28 | 2.22 | 16.14 | 13.99 |
| sgn-CI->fr | 0.69 | 0.37 | 9.5 | 0.01 | 4.4 | 0.3 | 0.71 | -1.08 | 9.7 | 0.52 | -2.23 | 14.8 |
| gss->el | 1.05 | 18.05 | 23.38 | 0.01 | 1.86 | 5.72 | 0.6 | 8.12 | 14.54 | 6.16 | 7.52 | 28.67 |
| sgn-MM->my | 4.5 | 44.03 | 27.65 | 0.06 | 8.75 | 9.7 | 1.88 | 9.6 | 16.89 | 7.64 | 41.13 | 27.41 |
| csq->hr | 0.01 | 22.21 | 3.32 | 0.07 | 21.07 | 8.89 | 0.08 | 22.5 | 8.16 | 0.1 | 20.28 | 11.73 |
| tza->sw | 0.03 | 20.47 | 7.76 | 0.06 | 21.66 | 10.24 | 0.22 | 18.46 | 14.05 | 0.21 | 21.73 | 15.58 |
| sgn-RS->sr | 0.02 | 21.65 | 3.9 | 0.07 | 21.54 | 7.26 | 0.09 | 22.56 | 8.1 | 0.16 | 22.35 | 11.55 |
| xml->ms | 0.06 | 35.9 | 12.56 | 0.67 | 46.52 | 15.58 | 0.42 | 46.99 | 18.58 | 0.41 | 41.08 | 15.89 |
| sfb->fr | 0.81 | 1.18 | 9.1 | 0.91 | 4.02 | 11.24 | 0.7 | -0.76 | 9.54 | 0.5 | -2.32 | 14.13 |
| nzs->en | 0.96 | 34.56 | 12.91 | 0.91 | 33.97 | 14.81 | 2.09 | 36.66 | 15.03 | 0.94 | 31.01 | 14.82 |
| dsl->da | 0.05 | 19.23 | 2.44 | 0.31 | 22.8 | 12.85 | 0.08 | 17.94 | 6.69 | 1.56 | 19.41 | 15.01 |
| hab->vi | 2.46 | 4.79 | 16 | 0.52 | 7.63 | 7.55 | 1.29 | 6.29 | 8.21 | 8.57 | -0.18 | 19.33 |
| nsl->no | 0.28 | 19.46 | 3.76 | 0.36 | 19.04 | 11.92 | 0.04 | 15.3 | 3.97 | 1.72 | 18.07 | 14.78 |
| isg->en | 1.2 | 36.38 | 13.52 | 0.02 | 26.33 | 4.31 | 1.68 | 37.95 | 15.28 | 1.38 | 32.07 | 15.24 |
| isr->he | 3.3 | 49.11 | 30.58 | 0.14 | 30.32 | 5.19 | 1.09 | 35.57 | 14.91 | 2.42 | 55.09 | 29.15 |
| ugy->es | 0.81 | 27.21 | 12.66 | 0.93 | 12.43 | 13.59 | 1.12 | 26.12 | 13.16 | 0.63 | 18.3 | 15.11 |
| jls->en | 0.78 | 34.97 | 13.86 | 1.27 | 34.65 | 14.84 | 1.04 | 36.14 | 16.12 | 0.68 | 32.17 | 14.82 |
| sgn-RW->rw | 0.13 | 14.83 | 6.88 | 0.03 | 13.59 | 4.95 | 0.23 | 9.07 | 11.07 | 0.09 | 9.48 | 10.54 |
| eth->am | 0.37 | 56.43 | 15.7 | 0.01 | 24.34 | 0.87 | 0 | 28.35 | 1.37 | 0.3 | 55.98 | 14.91 |
| sgg->de | 0.81 | 20.96 | 13.17 | 0.59 | 22.58 | 13.3 | 0.65 | 22.86 | 14.88 | 0.15 | 22.9 | 13.4 |
| ugn->en | 0.28 | 32.23 | 11.51 | 1.08 | 34.37 | 14.36 | 1.1 | 36.95 | 15.63 | 0.48 | 31.43 | 14.61 |
| sgn-PG->en | 1.13 | 35.64 | 13.8 | 1.61 | 32.9 | 16.58 | 1.74 | 36.21 | 15.6 | 0.93 | 32.72 | 16.07 |

Table 9: Evaluation Results on M91, CSIG, CSPO and MFT.

| Language Pair | M91 | | | CSIG | | | CSPO | | | MFT | | |
|---|---|---|---|---|---|---|---|---|---|---|---|---|
| | BLEU | BLEURT | chrF | BLEU | BLEURT | chrF | BLEU | BLEURT | chrF | BLEU | BLEURT | chrF |
| tsm–>tr | 1.17 | 17.41 | 10.77 | 0.01 | 9.4 | 7.26 | 0.09 | 21.78 | 11.47 | 0.48 | 12.74 | 8.9 |
| dse–>nl | 0.08 | 19.32 | 4.77 | 0.02 | 15.57 | 3.81 | 0 | 13.61 | 2.83 | 0.1 | 16.67 | 12.31 |
| vgt–>nl | 0.02 | 18.54 | 5.21 | 0.03 | 15.41 | 4.6 | 0.01 | 13.42 | 2.75 | 0.14 | 16.99 | 12.05 |
| lsl–>lv | 0.19 | 40.5 | 4.27 | 0.03 | 21.88 | 3.71 | 0.14 | 20.19 | 8.86 | 0.83 | 22.6 | 10.42 |
| eso–>et | 0.03 | 24.41 | 4.06 | 0.06 | 19.88 | 4.88 | 0.14 | 16.79 | 8.01 | 0.11 | 17.87 | 10.06 |
| aen–>hy | 3.04 | 56.72 | 27.18 | 0.03 | 29.38 | 5.81 | 0.1 | 32.28 | 11.06 | 0.2 | 57.51 | 13.82 |
| lls–>lt | 1.57 | 31.09 | 8.68 | 0.05 | 23.8 | 5.52 | 0.07 | 19.25 | 8.93 | 0.73 | 24.97 | 9.16 |
| nsp–>ne | 2.25 | 50.5 | 28.44 | 0.02 | 28.27 | 3.04 | 0.04 | 43.11 | 8.81 | 1.79 | 52.75 | 23.62 |
| bqn–>bg | 0.1 | 18.29 | 7.7 | 0.14 | 22.75 | 8.04 | 0.1 | 17.59 | 5.14 | 1.79 | 29.66 | 13.83 |
| hks–>zh-TW | 3.96 | 25.6 | 24.22 | 3.14 | 17.66 | 7.72 | 3.17 | 17.71 | 8.33 | 5 | 19.37 | 21.45 |
| sgn-SR–>nl | 0.03 | 17.68 | 4.66 | 0.01 | 15.76 | 2.95 | 0.01 | 12.3 | 2.68 | 0.12 | 18.79 | 13.44 |
| sqk–>sq | 1.01 | 34.96 | 13.84 | 0.02 | 21.64 | 4.51 | 0.09 | 18.54 | 8.97 | 0.35 | 26.98 | 8.08 |
| sgn-KH–>km | 2.87 | 49.28 | 32.52 | 0 | 22.58 | 0.19 | 0 | 20.62 | 0.94 | 2.84 | 47.95 | 28.47 |
| msr–>mn | 0.91 | 17.09 | 4.83 | 0.04 | 17.3 | 2.51 | 0.06 | 18.01 | 2.68 | 1.36 | 16.72 | 6.21 |
| asq–>de | 0.6 | 20.8 | 12.86 | 0.45 | 22.57 | 13.26 | 0.72 | 23.81 | 14.98 | 0.15 | 22.87 | 13.34 |
| sqs–>si | 2 | 53.36 | 25.02 | 0 | 27.95 | 0.03 | 0 | 37.97 | 5.32 | 4.61 | 55.27 | 29.44 |
| sgn-SI–>sl | 0.01 | 29.42 | 3.26 | 0.05 | 18.79 | 6.18 | 0.11 | 19.75 | 8.08 | 0.08 | 18.93 | 8.96 |
| sgn-BI–>fr | 0.43 | 5.56 | 7.9 | 0.18 | 5.7 | 0.16 | 1.03 | -1.39 | 9.9 | 1.29 | 0.09 | 14.99 |
| sgn-CM–>fr | 1.13 | 3.51 | 9.49 | 0.27 | 7.97 | 0.48 | 1.4 | -3.04 | 10.23 | 0.95 | -4.75 | 11.3 |
| sgn-FJ–>en | 0.34 | 41.56 | 13.94 | 0.69 | 33.64 | 16.89 | 0.72 | 35.4 | 15.42 | 0.94 | 30.54 | 16.08 |
| sgn-LB–>ar | 0.03 | 37.63 | 1.47 | 0.02 | 16.26 | 1.56 | 0.02 | 32.27 | 1.95 | 1.83 | 24.95 | 11.12 |

Table 10: continuation: Evaluation Results on M91, CSIG, CSPO and MFT.