# OpenReview forum: "JWSign: A Highly Multilingual Corpus of Bible Translations for more Diversity in Sign Language Processing"
_EMNLP/2023/Conference — EMNLP 2023 Findings_

### Official Review · Reviewer_Hyq8 · 2023-07-31

**Soundness:** 4

**Excitement:**

3: Ambivalent: It has merits (e.g., it reports state-of-the-art results, the idea is nice), but there are key weaknesses (e.g., it describes incremental work), and it can significantly benefit from another round of revision. However, I won't object to accepting it if my co-reviewers champion it.

**Paper Topic And Main Contributions:**

The paper proposes a compilation of the existing Jehovah’s Witnesses Bible sign language data sets as a new multilingual (98 sign languages) data set. The paper makes the first attempts at creating sign-to-text translation systems using the proposed dataset.

**Questions For The Authors:**

A. Why have you implemented a sign-to-text translation; when the dataset suggests the text-to-sign?

**Reasons To Accept:**

This greatly contributes to the sign language community; the work is relevant for cross-sign studies and translation evaluation.

The paper is clearly written, extensive statistics are given, and a lot of possible limitations and criticisms are addressed.

**Reasons To Reject:**

The proposed dataset was created with the following flow: original text to sign language translation. However, the common sign language dataset or corpus is created in reverse: original signing to textual annotations. That inconsistency has to be always stated. That makes the proposed dataset incomparable to other sign language corpora, created in a reversed way.

**Reproducibility:**

3: Could reproduce the results with some difficulty. The settings of parameters are underspecified or subjectively determined; the training/evaluation data are not widely available.

**Reviewer Confidence:**

2: Willing to defend my evaluation, but it is fairly likely that I missed some details, didn't understand some central points, or can't be sure about the novelty of the work.

---

> ### Author Rebuttal · Authors · 2023-08-29
>
> Q. Why have you implemented a sign-to-text translation; when the dataset suggests the text-to-sign?
>
> A. We made the choice to have our first baseline on sign-to-text translation because after consultation with some members of the deaf community, we realised it is the task with the most pressing practical usage right now (compared to text-to-sign). From the perspective of breaking down language barriers, sign-to-text is more important, because in the other direction speech recognition can be used as a substitute tool / solution for deaf people to understand what hearing people are saying.
>
> Q. However, the common sign language dataset or corpus is created in reverse: original signing to textual annotations. That makes the proposed dataset incomparable to other sign language corpora, created in a reversed way.
>
> A.
>
> It is not true that commonly, sign language datasets are created in the way you suggest  (signed language, such as ASL, first - spoken language translation, such as English, second). Almost all parallel datasets for sign languages are created by translating in the opposite direction.
>
> For example, this is true for 6 out of the 7 datasets we compared to in Table 1:
>
> PHOENIX ; speech (from TV Broadcast) → sign
>
> KETI ; text → sign
>
> CSL-Daily ; text → sign
>
> BOBSL ; speech (from TV Broadcast) → sign
>
> How2Sign ; text → sign
>
> OpenASL ; sign → text
>
> SP-10 ; text → sign
>
> JWSign (proposed dataset) ; text → sign
>
>
> It is also not true that this characteristic of a dataset (which language is the original one, and which is a translation) makes it incomparable to other datasets. If you think otherwise, we kindly ask you to provide some evidence (such as a previous paper) for this.
>
> We hope you are satisfied with our response and might reconsider your scores.

---

### Official Review · Reviewer_Ty6U · 2023-08-04

**Soundness:** 3

**Excitement:**

3: Ambivalent: It has merits (e.g., it reports state-of-the-art results, the idea is nice), but there are key weaknesses (e.g., it describes incremental work), and it can significantly benefit from another round of revision. However, I won't object to accepting it if my co-reviewers champion it.

**Justification For Ethical Concerns:**

It is unsure whether the dataset collection and usage are compliant with the terms of use of jw.org.

**Paper Topic And Main Contributions:**

The paper propose a new multilingual dataset for sign language processing, JWSign, based on Bible translations. The dataset includes 2530 hours of Bible translations in 98 sign languages.

The authors provided baseline evaluation results for bilingual and multilingual training.

**Reasons To Accept:**

The motivation is clear and the paper provides a nice background on the sign language datasets.

The proposed multilingual dataset has a higher diversity in terms of number of spoken languages, collected from ~1500 signers with diverse demographics, and is also significantly larger and longer compared to other datasets.

**Reasons To Reject:**

The licensing terms of the data and of the work don't seem to be described in the paper, which may lead to issues using it.

I have also some concerns about whether the baselines are strong enough. The results of the baselines seem quite low overall, and it could be good to have some more analysis about the reasons. The authors attached the detailed evaluation results in the Appendix, but it could be good to have more in-depth analysis, ex. about the variance of the results, whether there are similarities between the sign languages or if they are very different, whether there are error patterns?

**Reproducibility:**

3: Could reproduce the results with some difficulty. The settings of parameters are underspecified or subjectively determined; the training/evaluation data are not widely available.

**Reviewer Confidence:**

3: Pretty sure, but there's a chance I missed something. Although I have a good feel for this area in general, I did not carefully check the paper's details, e.g., the math, experimental design, or novelty.

**Typos Grammar Style And Presentation Improvements:**

Presentation:

- Would it be possible to adjust a bit the numbers in Table 4 so that they appear larger?
- Table 8: a separation rule is missing for M36

---

> ### Author Rebuttal · Authors · 2023-08-29
>
> Q. The results of the baselines seem quite low overall, and it could be good to have some more analysis about the reasons.
>
> A. Sign Language Translation (Sign-to-Text in this case) has proven to be a very challenging task and our baseline scores are in line with results of shared tasks as the WMT Shared Task on Sign Language Translation where BLEU scores are usually below 2 [1,2]. Additionally, when looking at the “Avg” column in Table 1, the average length of samples in our dataset is two or three times that of  other datasets. This is one of the reasons why other datasets such as PHOENIX, CSL-Daily or SP-10 usually have higher BLEU scores. Additionally, the Bible domain is a hard domain compared to other datasets with easier domain and smaller vocabulary such as weather forecast.
>
> Q. The authors attached the detailed evaluation results in the Appendix, but it could be good to have more in-depth analysis, ex. about the variance of the results, whether there are similarities between the sign languages or if they are very different, whether there are error patterns?
>
> A. Given that we had 98 language pair directions, our first analysis tried grouping them into fewer classes i.e. “High”, “Medium”, “Low”, “Very Low” and “Zero” and having a general overview of the trend in the scores across the board. With our 1 additional page, we intend to include a more fine-grained analysis, for instance on translation of Japanese Sign Language to Japanese, having the highest scores even with a low number of training samples.
>
> Q. Would it be possible to adjust a bit the numbers in Table 4 so that they appear larger?
>
> A. ​​Thank you, we will certainly fix this simple layout problem in the camera-ready version.
>
> Q. Table 8: a separation rule is missing for M36
>
> A. We kept this style to be consistent with Table 9 in the Appendix. We will find a way to better separate results visually in the camera-ready.
>
> [1] Findings of the First WMT Shared Task on Sign Language Translation (WMT-SLT22) (https://aclanthology.org/2022.wmt-1.71)
>
> [2] https://ocelot-wmt23.mteval.org/leaderboard/3
>
>
> We are happy to clarify any remaining questions you may have before reconsidering your scores. Thank you.

---

### Official Review · Reviewer_x6To · 2023-08-05

**Soundness:** 3

**Ethical Concerns:**

Yes

**Excitement:**

3: Ambivalent: It has merits (e.g., it reports state-of-the-art results, the idea is nice), but there are key weaknesses (e.g., it describes incremental work), and it can significantly benefit from another round of revision. However, I won't object to accepting it if my co-reviewers champion it.

**Justification For Ethical Concerns:**

The primary concern with the dataset is ethical issues. As the dataset is specific to a particular domain, it is challenging to validate the impact of the dataset in a negative direction. Moreover, the authors claim that the data strictly and entirely belongs to JW. It would be good to go through an ethical review to make it available for the data-driven natural language processing community.
Validating multicultural data for Sign languages may require more expertise.

**Paper Topic And Main Contributions:**

The paper curates a new multilingual corpus for sign language processing, using Bible Translations to capture more diversity in the dataset. The proposed dataset covers 98 sign languages with a varied number of samples for each of the sign languages ranging from 2 samples to 31k samples. The paper further provides baseline results over 36 bilingual pairs. Moreover, the paper also explores the multilingual setting for sign language processing and reports the findings.



**Questions For The Authors:**

* Did you consider brainstorming about the limited domain language being covered in the dataset? It would be good to highlight the ethical concerns in more detail with some efforts made to study this problem in more detail.


**Reasons To Accept:**

* The paper covers a wide range of sign languages and may act as a valuable resource for the sign language processing community to study the linguistic differences between various signed languages.

* The paper provides a detailed set of experiments over multiple combinations for translation. Though the translation scores are not high, it might still be usefull to study these mappings from one sign language to another.

* The paper clearly highlights the primary limitation of the proposed dataset of being in a limited domain, which may not be the true representative of communication in sign language.




**Reasons To Reject:**

* Though the dataset is diverse and unique in terms of multilingualism, the number of samples still remains a significant concern for the proposed dataset. It may be out of the scope of this work to collect more samples for these languages due to resource constraints. However, the usage of this dataset in the data-driven community will be limited for developing a reliable translation system.

* The primary concern with the dataset is ethical issues. As the dataset is specific to a particular domain, it is challenging to validate the impact of the dataset in a negative direction. Moreover, the authors claim that the data strictly and entirely belongs to JW. It would be good to go through an ethical review to make it available for the data-driven natural language processing community.



**Reproducibility:**

3: Could reproduce the results with some difficulty. The settings of parameters are underspecified or subjectively determined; the training/evaluation data are not widely available.

**Reviewer Confidence:**

2: Willing to defend my evaluation, but it is fairly likely that I missed some details, didn't understand some central points, or can't be sure about the novelty of the work.

---

> ### Author Rebuttal · Authors · 2023-08-29
>
> Q. Did you consider brainstorming about the limited domain language being covered in the dataset?It would be good to highlight the ethical concerns in more detail with some efforts made to study this problem in more detail
>
> A. Yes we did brainstorm about the limited domain language in this dataset. We dedicated Section 2.1. showcasing how data of the Bible domain has been extensively used in the advancements of NLP for languages in other modalities. A recent example is Meta’s Massively Multilingual Speech (MMS) model which was released last May. This work introduced a new dataset; MMS-lab dataset, containing Bible translations in 1,107 languages which helped achieve state of the art in multilingual speech-to-text translation. See “Our Approach” section https://about.fb.com/news/2023/05/ai-massively-multilingual-speech-technology/ ;
>
> “””
> Our Approach
> Collecting audio data for thousands of languages was our first challenge because the largest existing speech datasets cover 100 languages at most. To overcome this, we turned to religious texts, such as the Bible, that have been translated in many different languages and whose translations have been widely studied for text-based language translation research.
> By considering unlabeled recordings of various other Christian religious readings, we increased the number of languages available to more than 4,000. While this data is from a specific domain and is often read by male speakers, our analysis shows that our models perform equally well for male and female voices. And while the content of the audio recordings is religious, our analysis shows that this doesn’t bias the model to produce more religious language.
> “””
>
> This suggests that carefully using data from a religious domain does not bias to produce more religious language and instead, can be a crucial catalyst to reach state of the art for multilingual machine translation.
> Also, one other way to use this dataset will be to fine-tune smaller datasets of a more desirable domain on a pretrained model from this one. Given the extreme scarcity of sign language datasets compared to spoken languages in text or audio modalities, this dataset will be very useful to the sign language processing community.
>
> Finally, the multi-parallel nature of this dataset makes it the perfect candidate to help (sign) linguists study cross-sign linguistics patterns so as to have a better appreciation of sign languages.
>
> We are happy to clarify any remaining questions you may have before reconsidering your scores. Thank you.

---

### Meta-Review · Area_Chair_PVtZ · 2023-09-17

**Recommendation:** 4

**Metareview:**

The paper introduces a novel multilingual dataset for sign language processing, leveraging Bible Translations to enhance diversity. The paper also presents baseline results to bilingual pairs and explores the application of multilingual approaches in sign language processing. In general the paper seems like a good contribution, but there might be some issues involving the data licensing that needs to be addressed.

---

### Decision · Program_Chairs · 2023-10-07

**Decision:**

Accept-Findings

**Comment:**

The paper introduces a novel multilingual dataset for sign language processing, leveraging Bible Translations to enhance diversity. The paper also presents baseline results to bilingual pairs and explores the application of multilingual approaches in sign language processing. In general the paper seems like a good contribution, but there might be some issues involving the data licensing that needs to be addressed.